# Quantitative Analysis of Photothermal Therapy of Tumor Tissue Using Various Gold Nanoparticle Injection Schemes

**DOI:** 10.3390/pharmaceutics15030911

**Published:** 2023-03-10

**Authors:** Donghyuk Kim, Hyunjung Kim

**Affiliations:** Department of Mechanical Engineering, Ajou University, Suwon-si 16499, Republic of Korea

**Keywords:** apoptosis, direct injection, gold nanoparticles, photothermal therapy, thermal damage

## Abstract

Photothermal therapy is a new chemotherapy technique using photothermal effects, a phenomenon in which light energy is converted into thermal energy. Since the treatment technique is performed without surgical incision, it does not cause bleeding and patients are expected to make rapid recoveries, which are significant advantages. In this study, photothermal therapy with direct injection of gold nanoparticles into tumor tissue was simulated through numerical modeling. The treatment effect resulting from changing the intensity of the irradiated laser, volume fraction of the injected gold nanoparticles, and number of gold nanoparticle injections was quantitatively evaluated. The discrete dipole approximation method was applied to calculate the optical properties of the entire medium, and the Monte Carlo method was applied to identify the absorption and scattering behavior of lasers in tissue. In addition, by confirming the temperature distribution of the entire medium through the calculated light absorption distribution, the treatment effect of photothermal therapy was evaluated, and the optimal treatment conditions were suggested. This is expected to accelerate the popularization of photothermal therapy in the future.

## 1. Introduction

Photothermal therapy is a new chemotherapy technique that uses the photothermal effect [1], a phenomenon in which light energy is converted into thermal energy, to increase the temperature of tumor tissues, killing them [2,3]. Since the treatment is performed without incising the affected area, it has the advantages of no bleeding and rapid recovery compared to conventional treatment techniques through incision [4,5,6]. Light energy, in the form of a laser, is irradiated onto the surface of the skin. Since the energy of the laser cannot be transmitted to great depths, this method has mainly been restricted to research on skin cancer [7,8].

Photothermal therapy primarily utilizes lasers in the near-infrared (NIR) portion of the electromagnetic spectrum. This is because the depth of energy penetration in this range is greater than that of lasers in other wavelength bands. Furthermore, the light absorption coefficient of the medium itself is low in this region, which can reduce unnecessary thermal damage to surrounding tissues [9,10,11]; however, this also means that NIR irradiation cannot raise the temperature of the medium to the point that induces tumor tissue death. To compensate for this, treatment is performed by administering a light absorption enhancer to tumor tissue to increase the medium’s light absorption coefficient at a specific wavelength [12,13,14]. To achieve this, the light absorption enhancer uses localized surface plasmon resonance (LSPR), a phenomenon that induces collective oscillation of electrons at a specific wavelength [15,16]. Accordingly, it is possible to raise the temperature of the light-absorption enhanced tissue to the range required for biological tissue death, even if a laser in the NIR region is used. Light absorption enhancers are classified according to their constituent materials; among them, gold nanoparticles (GNPs), with excellent biocompatibility, are most common [17,18,19].

As described above, photothermal therapy aims to kill tumor tissue through temperature increase. To this end, it is first necessary to verify the temperature ranges at which the forms of death of various types of biological tissue occur. In general, the death of biological tissue according to temperature is divided into apoptosis and necrosis [20,21,22]. Apoptosis is also called cell suicide, and refers to a form of self-destruction which occurs without damaging surrounding tissues. Necrosis, however, involves the destruction of the intracellular contents in the form of leakage and has the possibility of cancer metastasis. Apoptosis is known to be expressed mainly at 43–50 °C, while necrosis is known to occur at higher temperatures. For this reason, it is an important issue in photothermal therapy to maintain the temperature range of apoptosis expression by controlling the appropriate laser conditions.

Photothermal therapy is currently being studied from various perspectives. Muzzi et al. [23] proposed the use of particles with an Fe_3_O_4_ shell structure on a star-shaped Au core to increase the light conversion efficiency of the light absorption enhancer. Magnetometry and magneto-optical spectroscopy reportedly revealed a pure magnetite shell with excellent saturation magnetization. Furthermore, the plasmonic resistance in the Au@Fe_3_O_4_ system could be converted to 640 nm, showing applicability for photothermal therapy and visible optical imaging. A preliminary experiment was conducted to irradiate a laser on a cancer cell culture suspension into which Au@Fe_3_O_4_ was injected, and its suitability for optical response and photothermal therapy was confirmed. Bi et al. [24] proposed Ag_2_O_2_ nanoparticles, which can control the release of reactive oxygen species, to reduce the toxic side effects of metal peroxide nanoparticles designed to increase oxidative stress. In vitro and in vivo experiments confirmed that Ag_2_O_2_ nanoparticles had a mortality efficiency of 99.9999% or more within 10 min, provided improved antibacterial and antibiotic membrane functions, and accelerated wound suture against Staphylococcus aureus infection with excellent cell and blood compatibility. Through this, a high-efficiency, non-invasive, and safe treatment method for combating bacterial infectious diseases was presented. Wang et al. [25] analyzed the effect of photothermal therapy on tumors containing gold nanoparticles through numerical analysis. The temperature distribution of tumors generated from the skin surface and the surrounding normal tissues was confirmed using the Monte Carlo method, and analysis was performed by varying the treatment conditions for photothermal therapy, such as the tumor size, irradiating laser intensity, and volume fraction of GNPs in the tumor. In addition, the thermal damage to the tissue was confirmed through the Arrhenius thermal damage integral. Yin et al. [26] studied photothermal therapy inside the tissue through numerical analysis. The effects of laser intensity, volume fraction of GNPs, anisotropic scattering characteristics of nanoparticles, and laser incident angle were investigated. Based on this, various treatment strategies, such as single heat source and multiple heat sources, were also identified. In the case of single-dose treatment, it was confirmed that there was almost no difference for the various laser incident angles; in the case of the split treatment, better treatment effects were observed when the laser was irradiated at a constant angle than in the vertical direction. In addition, it was confirmed that multiple heat sources had better therapeutic effects than a single heat source. Guglielmelli et al. [27] developed keratin-coated gold nanoparticles (Ker-AuNPs) as a highly efficient photosensitizing therapeutic agent. The physical, photothermal, chemical, and morphological properties of Ker-AuNPs were investigated by various methods, including dynamic light scattering, ζ-potential, Fourier transform infrared spectroscopy, and X-ray photoelectron spectroscopy. In addition, in vitro experiments were conducted on human glioblastoma cell lines to confirm the efficient cellular absorption, good biocompatibility and local photothermal heating of Ker-AuNPs. Annesi et al. [28] studied an antimicrobial methodology based on gold nanorods (GNRs). The anti-microbial effect of GNRs on Escherichia coli bacteria was confirmed, and it is important to control the concentration of GNRs to exclude toxic effects on cells and to generate the amount of heat required to raise the temperature to 50 degrees in about 5 min in the near-infrared region. In addition, as a result of the experiment, it was confirmed that killing efficiency suitable for reducing Escherichia coli populations to about 2 log colony-forming units was achieved. Candreva et al. [29] synthesized spherical GNPs with a diameter of 50 nm coated with polyethylene glycol and administered into Escherichia coli cultures to activate plasmon in the visible light region. Experiments were performed in the dark and under laser irradiation, with varying concentrations of GNPs. In the dark, 46% of bacterial growth was inhibited, while laser irradiation at the same concentration resulted in 99% growth inhibition. This was attributed to the fact that the bacterial wall promotes the formation of light-induced clusters of nanoparticles, resulting in an increase in temperature and a bactericidal effect. Furthermore, this photothermal effect is achieved with low intensity laser irradiation only when the pathogen is present, proving that this is an innovative response system to bacterial infections.

Summarizing the preceding studies, research on photothermal therapy under various conditions has been conducted through experiments and numerical analysis. However, although photothermal therapy is a treatment technique based on the heat transfer phenomenon, existing research methods have derived results based on simple phenomenological observations through in vitro and in vivo experiments or quantitative rate of apoptosis, which are commonly used in the biological field. In addition, in the case of thermal damage, only the presence of damage was determined through the Arrhenius thermal damage integral, and it was assumed that GNPs were uniformly distributed in the tumor tissue. There are two main methods of injecting GNPs into tumor tissue: direct injection and intravenous injection [30,31]. Direct injection has the advantage that most of the injected GNPs are located in the tumor tissue; however, the location where they can be injected is limited, and GNPs tend to be concentrated around the injection point. Although intravenous injection allows GNPs to be injected at all locations where blood vessels pass, GNPs are lost in the process of being delivered to the desired location. In the case of direct injection of GNPs into the tumor, the GNPs may have loss from the tumor environment by various immunity mechanisms, but there seems to be less loss of GNPs than in cases of intravenous injection [32].

In this study, numerical analysis was used to study photothermal therapy that simulates the direct injection of GNPs into tumor tissue generated from the skin surface. The target tumor was selected as squamous cell carcinoma, and the four skin layers were implemented through numerical modeling. Analysis was conducted to determine the effect of changing the intensity of the laser and the volume fraction of the injected GNPs, which are some of the variable treatment conditions of photothermal therapy. In addition, for given GNP volume fractions, the effect of varying the number of injections on the effectiveness of photothermal therapy was quantitatively evaluated through the apoptotic variable suggested by Kim et al. [33]. Based on these results, the conditions that produce the optimal photothermal therapy effect were determined for the given treatment situation.

## 2. Material Properties and Numerical Methods

### 2.1. Optical Properties of Gold Nanoparticles and Biological Tissues

As mentioned above, photothermal therapy mainly utilizes lasers in the NIR region, and GNPs are injected into the tumor tissue to compensate for the low light absorption coefficient of the tissues in this region. GNPs increase light absorption at a specific wavelength (in this case inside the NIR band) due to the localized surface plasmonic resistance (LSPR) phenomenon.

The optical properties of GNPs were calculated using the discrete dipole approximation (DDA) method [34,35]. The DDA method is a technique capable of calculating the absorption and scattering efficiency of nanoparticles of various shapes. Given a particular shape, the optical efficiency of the particle is obtained by calculating its electromagnetic properties, assuming that the dipoles are located at regular intervals. This technique compensates for the disadvantage that Mie theory can only calculate the shape of a sphere or an ellipse [36].

The first step of the DDA method is calculation of the polarization vector (*P*), determined through the interaction of the dipole and the local electric field (*E*) (Equation (2)), as per Equation (1). Here, α and *r* represent the polarizability and position vectors, respectively. In Equation (3), *E_inc,i_* represents the initial electric field and *k* represents the wave number. Equation (4) represents the interaction matrix between dipoles. Here, *r_ij_* represents $\left|r_{i}-r_{j}\right|$, and *A_ij_* is an interaction matrix under the condition that *i* ≠ *j*. If *i* and *j* are equal, the interaction matrix can be simplified to $\alpha_{i}^{-1}$.
(1)$$P_{i}=\alpha_{i}\cdot E_{i}\left(r_{i}\right)$$
(2)$$E_{i}\left(r_{i}\right)=E_{inc,i}-\sum_{j\neq i}^{N}A_{ij}\cdot P_{j}(i,j=1,2,3,\ldots ,N)$$
(3)$$E_{inc,i}=E_{0}e^{i(k\cdot r_{i})}$$
(4)$$A_{ij}\cdot P_{j}=\frac{e^{i\left(k\cdot r_{ij}\right)}}{r_{ij}^{3}}\left\{k^{2}r_{ij}\times \left(r_{ij}\times P_{j}\right)+\frac{1-ikr_{ij}}{r_{ij}^{2}}\times \left[k^{2}P_{j}-3r_{ij}\left(r_{ij}\cdot P_{j}\right)\right]\right\}(i\neq j)$$

The optical cross section *C* can be calculated as shown in Equations (5)–(7), where * is a complex conjugate symbol.
(5)$$C_{abs}=\frac{4\pi k}{{\left|E_{0}\right|}^{2}}\sum_{i=1}^{N}\left\{Im\left[P_{i}\cdot {\left({\alpha_{i}}^{-1}\right)}^{*}P_{i}^{*}\right]-\frac{2}{3}k^{3}P_{i}P_{i}^{*}\right\}$$
(6)$$C_{ext}=\frac{4\pi k}{{\left|E_{0}\right|}^{2}}\sum_{i=1}^{N}Im(E_{inc,i}^{*}\cdot P_{i})$$
(7)$$C_{sca}=C_{ext}-C_{abs}$$

Finally, each optical efficiency *Q* can be calculated using Equation (8). Here, *V* and *r_eff_* and *V* are the volume and effective radius of the particle, respectively.
(8)$$Q_{abs}=\frac{C_{abs}}{\pi r_{eff}^{2}},Q_{ext}=\frac{C_{ext}}{\pi r_{eff}^{2}},Q_{sca}=\frac{C_{sca}}{\pi r_{eff}^{2}}$$
(9)$$r_{eff}={\left(\frac{3V}{4\pi }\right)}^{\left(1/3\right)}$$

Each optical efficiency calculated in this way is used in the final optical property calculation. The optical properties of GNPs are obtained from the volume fraction of GNPs (*f_v_*), the effective radius of the particle (*r_eff_*), and the optical efficiency *Q,* calculated through the DDA method as shown in Equation (10) [37]. After the optical properties of the GNPs are calculated, the optical properties of the entire medium with GNPs injected can be calculated as the sum of the optical properties of the GNPs and the optical properties of the medium, as shown in Equation (11) [38].
(10)$$\mu_{abs,np}=0.75f_{v}\frac{Q_{abs,np}}{r_{eff}},\mu_{sca,np}=0.75f_{v}\frac{Q_{sca,np}}{r_{eff}}$$
(11)$$\mu_{abs}=\mu_{abs,np}+\mu_{abs,m},\;\mu_{sca}=\mu_{sca,np}+\mu_{sca,m}$$

### 2.2. Validation of Numerical Process

In this study, an in vivo experiment using a BALB/c mouse was conducted to verify the proposed numerical analysis modeling. BALB/c mice are an inbred group among laboratory mice and are the most frequently used animals in research fields regarding tumors, inflammation, and autoimmunity. The experiment was conducted in the clean area of Ajou University’s Experimental Animal Center and was approved by the Institutional Biosafety Committee (IBC) (No. 2021-0034) and the Institutional Animal Care and Use Committee (IACUC) (No. 2021-0079). Female BALB/c mice with an age of 8 weeks and a weight of 15–20 g, purchased from ORIENT BIO Inc., were used, and experiments were performed while varying the input dose of GNPs. To develop skin cancer in BALB/c mice, DBMA-TPA two-stage skin carcinogenesis model was applied in this study [39]. The fur of the experimental mouse was shaved 2 days before the onset of carcinogenesis, and 50 μg of 7,12-dimethylbenz(α)anthracene (DMBA) was applied once to the back of the experimental mouse where light was blocked for the onset of carcinogenesis. After 2 weeks of application, 5 μg of 12-O-tetradecanoylphorbol-13-acetate (TPA) was applied to the mouse’s back twice a week. Both DMBA and TPA were dissolved in 200 μL of acetone before being applied, and shaving and reagent application were performed with the mouse anesthetized by a respiratory anesthesia machine using isoflurane.

Figure 1 is a schematic of the in vivo experimental for laser irradiation. The laser used in this study is the Cobolt 04-01 Series ‘Rumba’ model, sold by Cobolt. A 1064 nm single-wavelength laser with an intensity of 0.4 W and a diameter of 1 mm was used, with the beam diameter increased to 10 mm through a beam expander. In addition, an optical mirror was used to change the path of the beam from horizontal to vertical. Since the laser used in this study is in the infrared region, an IR viewer was used to verify the path of the laser beam.

Figure 2a,b shows the results on a mouse when laser was irradiated for 5 min on the skin cancer site after injecting 25 ug (*f_v_* ≈ 3 × 10^−4^) and 50 ug (*f_v_* ≈ 6.5 × 10^−4^) of GNPs, respectively. GNPs used in this study are a rod type shape with an effective radius of 10 nm and an aspect ratio of 6.7. They are dispersed at a concentration of 35 ug/mL in H_2_O. GNPs were purchased from Sigma Aldrich and the absorption spectrum of the nanoparticles shows a longitudinal peak at 1064 nm and a longitudinal absorbance OD of 1. As shown in the figure, there were blisters (red circles) on the mice’s skin due to burns in the affected area after laser irradiation compared to none before. These were classified as blisters on burns, known to be expressed at about 50–70 °C, as announced by Stoll et al. [40] and Leach et al. [41]. 

Based on the results of the in vivo experiments, the numerical analysis model proposed in this study was verified. The diameter of the laser was set to 10 mm as in the experiment, and the intensity was set to 0.4 W. The thermal and optical properties of BALB/c mice are summarized in Table 1. Figure 3 shows the results of numerical analysis under the same conditions as in the vivo experiment. The numerical analysis indicated a maximum temperature of 63.3 °C when 50 ug (*f_v_* ≈ 6.5 × 10^−4^) of GNPs were injected, with an overall temperature increase to above 50 °C in the irradiated part. Through this, it can be judged that the results derived from the in vivo experiment and the numerical analysis are similar. In addition, verification of the numerical model was performed through in vitro experiments using biomimetic phantoms. Comparison between the numerical analysis modeling and the experiment yielded an RMSE of the temperature change over time as an average of 0.1677. It was thus determined that the numerical analysis modeling was valid [33]. 

### 2.3. Numerical Investigation

In this study, the skin implemented in numerical analysis modeling was composed of four layers, including a squamous cell carcinoma generated from the skin surface. As shown in Figure 4, the squamous cell carcinoma with a radius of 2 mm and a depth of 2 mm was located inside a cylindrical skin layer with a radius of 15 mm and a depth of 20 mm. In order to achieve an appropriate laser penetration depth for the target skin cancer [44], A 1064 nm single-wavelength Gaussian distribution laser with a radius of 2 mm was used as the heat source with the same radius as the tumor. The thickness and thermal and optical properties of each skin layer and tumor tissue are summarized in Table 2.

In this study, it was assumed that GNPs in the tumor were not uniformly distributed but that GNPs were divided several times and directly injected, resulting in uneven distribution of GNPs inside the tumor. However, in the actual situation, after the GNPs are administered to the tumor tissue, the GNPs are not present only at that site due to various mechanisms. In this study, it was assumed that the treatment was performed immediately after the direct injection of GNPs, and GNPs were present in all the places where they were injected. It was assumed that GNPs were distributed in a spherical shape at each injection site, and the total volume fraction of GNPs in the tumor was assumed to be the same regardless of the number of injections of GNPs. For example, assuming that the initial input volume is 100, it is assumed that when the input is divided 4 times, it is divided into 25 volumetric units in each division. Therefore, the diffusion radius of individual GNPs injection decreases as the number of injections increases, as shown in Figure 5.

Table 3 shows the numerical analysis conditions selected in this study. The laser irradiation time was set to 600 s, and the intensity of the laser was increased in 2 mW intervals from 0–100 mW. The volume fraction of injected GNPs was increased from 10^−1^–10^−8^ at intervals of 10^−1^ in a total of eight steps. It was assumed that GNPs were distributed in the form of spheres with a diameter of 2 mm for all volume fractions based on one-time injection.

The GNPs used in the numerical analysis were rod-type nanoparticles with an aspect ratio of 6.67 and an effective radius of 20 nm. The GNPs utilized in the in vivo experiments have an effective radius of 10 nm, but the authors’ previous study confirmed that GNPs with an effective radius of 20 nm have the highest absorption efficiency in the rod-type [52]. Therefore, GNPs with an effective radius of 20 nm were utilized in this numerical analysis. It was assumed that the same GNPs were used in all cases, and the wavelength of the irradiating laser was fixed at 1064 nm. Calculated by the DDA method for these particles, the absorption efficiency at 1064 nm is 14.878 and the scattering efficiency is 3.1416. These were applied to Equation (10) to calculate the optical coefficients for different volume fractions. The optical properties of tissues including GNPs were calculated as the sum of the optical properties of the GNPs and those of the medium, calculated through the DDA method, and are summarized in Table 4.

To calculate the temperature distribution inside the tissue, the absorption distribution of the irradiated laser energy must first be calculated. In this study, a Monte Carlo technique that can consider both absorption and scattering behavior was used to calculate the distribution of light energy absorbed in the medium [53]. The Monte Carlo method is a technique that calculates probabilistically the degree of absorption and scattering of laser particles inside the medium using random numbers. Once the light absorption distribution is calculated, the temperature distribution in the medium can be computed through the explicit finite element method based on the thermal diffusion equation, as shown in Equation (12). Here, τ, ρ, and *c_v_* represent the time, density, and specific heat, respectively, while *F* and *P_l_* represent the fluence rate and laser intensity, respectively [54].
(12)$$∆T=\frac{∆\tau }{\rho c_{v}}\left(\begin{array}{c} \mu_{a}FP_{l}+\left(T_{x^{-}}-T\right)\frac{2k_{m}k_{{m,x}^{-}}}{k_{m}+k_{{m,x}^{-}}}\frac{1}{dx^{2}}+\left(T_{x^{+}}-T\right)\frac{2k_{m}k_{{m,x}^{+}}}{k_{m}+k_{{m,x}^{+}}}\frac{1}{dx^{2}}\\ +\left(T_{y^{-}}-T\right)\frac{2k_{m}k_{{m,y}^{-}}}{k_{m}+k_{{m,y}^{-}}}\frac{1}{dy^{2}}+\left(T_{y^{+}}-T\right)\frac{2k_{m}k_{{m,y}^{+}}}{k_{m}+k_{{m,y}^{+}}}\frac{1}{dy^{2}}\\ +\left(T_{z^{-}}-T\right)\frac{2k_{m}k_{{m,z}^{-}}}{k_{m}+k_{{m,z}^{-}}}\frac{1}{dz^{2}}+\left(T_{z^{+}}-T\right)\frac{2k_{m}k_{{m,z}^{+}}}{k_{m}+k_{{m,z}^{+}}}\frac{1}{dz^{2}} \end{array}\right)$$

Finally, the various conditions for performing photothermal therapy were varied, and the resulting treatment effect for squamous cell cancer was confirmed numerically. Information on how much apoptosis in tumor tissue was maintained, and the thermal damage of surrounding normal tissues was quantitatively confirmed to identify the optimal treatment conditions.

## 3. Results and Discussion

### 3.1. Temperature Distribution of Biological Tissues

Figure 6 shows the temperature change over time at the center of the tumor surface with respect to the number of injections and the volume fraction of GNPs (*f_v_*) for lasers of 40 mW intensity. Figure 6a shows the results for an *f_v_* of 10^−6^, while Figure 6b shows the results when GNPs were deposited in six separate injections, according to volume fractions. As shown in Figure 6a, the temperature of the center of the tumor surface varies according to the number of injections. In particular, when the GNPs were injected separately, excessive temperature rise did not occur, compared to the case in which GNPs were injected once in the center. As shown in Figure 6b, the temperature increase at the center of the tumor surface decreases as the *f_v_* decreases. Through this, it was confirmed that the temperature of the entire medium increased differently according to the various number of injected GNPs and *f_v_*. Accordingly, in the results described later, the temperatures of all points in the medium were checked, and the degree of maintenance of the temperature range where apoptosis occurred and the amount of thermal damage to surrounding normal tissues were quantitatively evaluated.

### 3.2. Apoptosis Occurrence Amount in Tumor Tissue

In this study, the apoptosis retention ratio ($\theta_{A}^{*}$) proposed by Kim et al. [29] was used to quantitatively identify how effectively apoptosis occurrence was maintained in the tumor tissue. The term $\theta_{A}^{*}$ is the average ratio of the tumor volume corresponding to the apoptosis temperature range of 43–50 °C to the total volume of the tumor at each time step (i.e., to obtain the average, the ratios at each time step are summed and the result is divided by the total treatment time). The maximum value of $\theta_{A}^{*}$ is 1, indicating that the occurrence temperature range of apoptosis is maintained for the entire treatment time at all locations in the tumor. Using this method, it is possible to quantitatively determine how much the apoptosis occurrence temperature range is maintained inside the tumor tissue during the total treatment time.

Figure 7 shows $\theta_{A}^{*}$ for each laser intensity *P_l_* according to the various *f_v_*. The tendency of $\theta_{A}^{*}$ is similar between various numbers of injections, so only the results of the three and six GNP injection cases are replicated in the graphs below. It was confirmed that there existed a *P_l_* corresponding to the maximum $\theta_{A}^{*}$ for each *f_v_* and for each number of injections. In addition, it was confirmed that as *f_v_* decreased, the *P_l_* corresponding to a maximum $\theta_{A}^{*}$ increased. This is because a decrease in *f_v_* results in reduced heat absorption capability of the medium for the laser, so a higher *P_l_* is needed to maintain the desired temperature range. As the number of injections increased, the *P_l_* corresponding to a maximum $\theta_{A}^{*}$ slightly decreased. This is because, as the number of injections increases, the injected GNPs are distributed more widely and evenly, so the tumor tissue is heated in a larger area even at a relatively low *P_l_*.

Figure 8 shows $\theta_{A}^{*}$ for each *P_l_* according to various number of injected GNPs for a given *f_v_*. Figure 8a shows the change in $\theta_{A}^{*}$ when *f_v_* is 10^−3^. In this case, there was a specific *P_l_* corresponding to a maximum $\theta_{A}^{*}$ for each number of injections. In addition, it was confirmed that the maximum value of $\theta_{A}^{*}$ was higher when GNPs were injected in two or more divided doses compared to one dose. As mentioned above, this is attributed to the increased GNPs diffusion radius which results in a more widespread temperature rise inside the tumor. Figure 8b shows the result when *f_v_* is 10^−8^, and it is confirmed that $\theta_{A}^{*}$ according to *P_l_* is similar for all numbers of injections. This is because the *f_v_* is very low and the effect of the GNPs’ injection is insignificant—the GNP injections do initiate an increase in the light absorption coefficient of the entire medium. Evaluating the overall treatment conditions, it was concluded that apoptosis occurrence temperature in tumor tissue was best maintained when the number of injections was five times, *f_v_* was 10^−5^, and *P_l_* was 58 mW.

### 3.3. Thermal Damage of Surrounding Normal Tissues

Previously, the degree to which the apoptosis occurrence temperature was maintained inside the tumor tissue was evaluated through $\theta_{A}^{*}$. However, this variable can only identify whether the phenomenon occurs inside the tumor tissue and cannot confirm the amount of thermal damage to the surrounding normal tissue. Accordingly, in this study, the amount of thermal damage to the surrounding normal tissue surrounding the tumor was quantitatively evaluated through the thermal hazard retention value ($\theta_{H}^{*}$) [29]. The variable $\theta_{H}^{*}$ weights the degree to which the phenomena occurred in normal biological tissues according to various temperature ranges, and then averages the weighted sums at each time step for all points in the normal tissues surrounding the tumor tissue. The minimum value of $\theta_{H}^{*}$ is 1, which indicates that no thermal damage occurs in the surrounding normal tissue, and the value of $\theta_{H}^{*}$ increases as the thermal damage increases. In this study, only normal tissues from the tumor tissues’ end to 50% of the tumor tissue diameter were considered for thermal damage analysis.

Figure 9 shows $\theta_{H}^{*}$ for each *P_l_* according to *f_v_*. Figure 9a represents the case of three GNPs injections in the tumor tissue, while Figure 6b represents six injections. In both cases, as *f_v_* decreases, the amount of thermal damage to surrounding normal tissues decreases. This is because as *f_v_* decreases, the temperature increase of the tumor tissue occurs less, as shown in Section 3.2.

Figure 10a,b shows $\theta_{H}^{*}$ for each *P_l_* according to the number of GNPs injections when *f_v_* is 10^−3^ and 10^−8^, respectively. As shown in Figure 10a, as the number of injections increases, so does the amount of thermal damage in the surrounding normal tissue. This is because as the number of injections increases, the diffusion radius of GNPs within the tumor increases, resulting in a temperature rise across a wider range of the tumor tissue, and, as a result, more laser heat absorbed from the tumor tissue is transferred to the surrounding normal tissue.

### 3.4. Quantitative Analysis of Photothermal Therapy Effect

Section 3.2 and Section 3.3 evaluated the degree of maintenance of apoptosis occurrence inside the tumor tissue and the amount of thermal damage to the surrounding normal tissue, respectively. However, since the two phenomena do not occur individually, but rather simultaneously, they should be considered in combination. Therefore, this study identified treatment conditions that maximize the occurrence of apoptosis in tumor tissue and minimize thermal damage to surrounding normal tissues, which is the goal of photothermal therapy. This was accomplished using the effective apoptosis retention ratio ($\theta_{eff}^{*}$) [29], defined as the ratio of $\theta_{A}^{*}$ and $\theta_{H}^{*}$ calculated above. The maximum value of $\theta_{eff}^{*}$ is 1, corresponding to a case where there is no thermal damage at all to the surrounding normal tissue while all points inside the tumor tissue maintained the apoptosis temperature range for all time steps.

Figure 11 shows the resulting $\theta_{eff}^{*}$ at each *P_l_* according to various conditions of photothermal therapy. The trend of the overall graph was derived in the same way as $\theta_{A}^{*}$, and it was confirmed that there was a *P_l_* corresponding to an optimal therapeutic effect for each *f_v_* and number of GNPs injections. At *P_l_* values below the optimum point, the temperature of the tumor tissue did not rise sufficiently, resulting in a sub-optimal treatment effect. At values of *P_l_* greater than the optimum point, the temperature of the tumor tissue increased excessively, resulting is significant transfer of laser heat to the surrounding normal tissue, increasing the thermal damage. When summarizing the effects of all treatment conditions, it was confirmed that the optimal treatment effect was achieved when the number of injections was six, *f_v_* was 10^−3^, and *P_l_* was 42 mW. The optimal *P_l_* for treatment was lower than that for $\theta_{A}^{*}$; this is because a higher *P_l_* results in greater heat transfer from the tumor tissue to the surrounding normal tissue, increasing the amount of thermal damage. Likewise, the higher optimal *f_v_* for treatment vs. $\theta_{A}^{*}$ is explained as follows: the effect of *f_v_* on heat absorption (caused by the change in the light absorption coefficient) is greater than the effect of laser penetration depth (caused by higher *P_l_*) on the maximum heat absorption depth. Through this analysis, the treatment conditions that produce optimal treatment effects were identified for the case of photothermal therapy on a skin layer where squamous cell carcinoma has occurred.

## 4. Conclusions

In this study, a skin layer in which squamous cell carcinoma occurred was simulated with numerical modeling to evaluate the effect of photothermal therapy under various conditions. To simulate the actual treatment situation, it was assumed that gold nanoparticles were directly injected and partially distributed inside the tumor, and numerical analysis was performed while changing the intensity of the irradiated laser, the volume fraction of the injected gold nanoparticles, and the number of direct injections. To consider the optical effect when gold nanoparticles were administered to tumor tissue, the optical properties of the tissue were calculated by applying the discrete dipole approximation method. In addition, the Monte Carlo method was used to analyze the absorption and scattering behavior of light energy in the medium, and the temperature distribution in the medium was calculated based on the thermal diffusion equation.

Using the obtained temperature distribution, the degree of maintenance of apoptosis occurrence temperature in tumor tissue was derived through the apoptosis retention ratio, and the thermal damage of surrounding normal tissues was calculated through the thermal hazard retention ratio. In addition, to analyze the therapeutic effect of actual treatment conditions, the photothermal therapy conditions that produced optimal therapeutic effects were evaluated through the effective apoptosis retention ratio, which balanced the two ratios above. For the tumor tissue conditions presented in this study, it was concluded that the optimal treatment effect was achieved when the volume fraction of gold particles was 10^−3^, deposited over six direct injections and irradiated at a laser power of 42 mW. It is thus judged that it is possible to present a reference point for optimal treatment in photothermal therapy in which GNPs are directly injected into the tumor. Furthermore, it is believed that the popularization of photothermal therapy can be accelerated by optimizing treatment conditions from a clinical perspective through the process of performing actual treatment trials. In addition, it is necessary to study a more realistic situation, reflecting the phenomenon that GNPs are dissipated by various mechanisms immediately after they are injected.

## Figures and Tables

**Figure 1 pharmaceutics-15-00911-f001:**
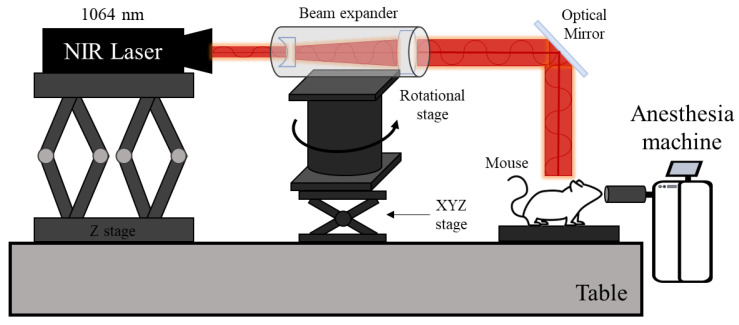
Schematic of in vivo experiment.

**Figure 2 pharmaceutics-15-00911-f002:**
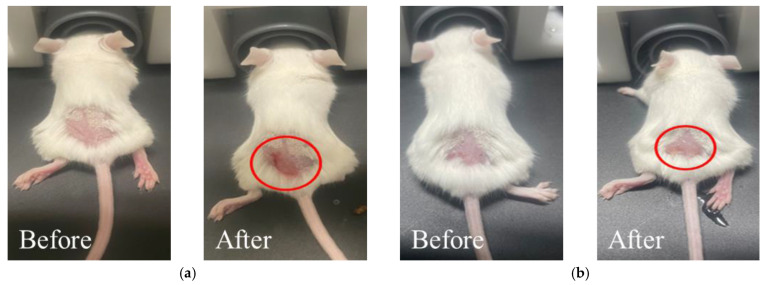
Validation experiment results: (**a**) 25 μg dose, irradiated for 5 min; (**b**) 50 μg dose, irradiated for 5 min.

**Figure 3 pharmaceutics-15-00911-f003:**
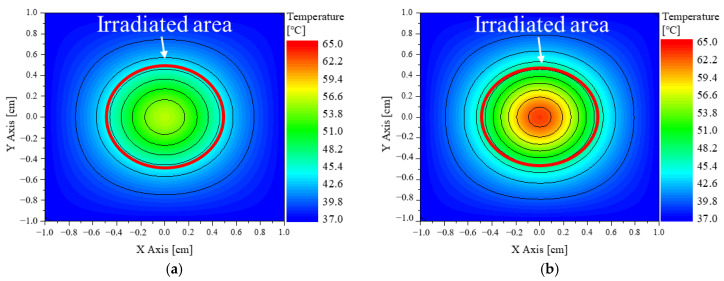
Validation numerical analysis results: (**a**) 25 μg dose, irradiated for 5 min; (**b**) 50 μg dose, irradiated for 5 min.

**Figure 4 pharmaceutics-15-00911-f004:**
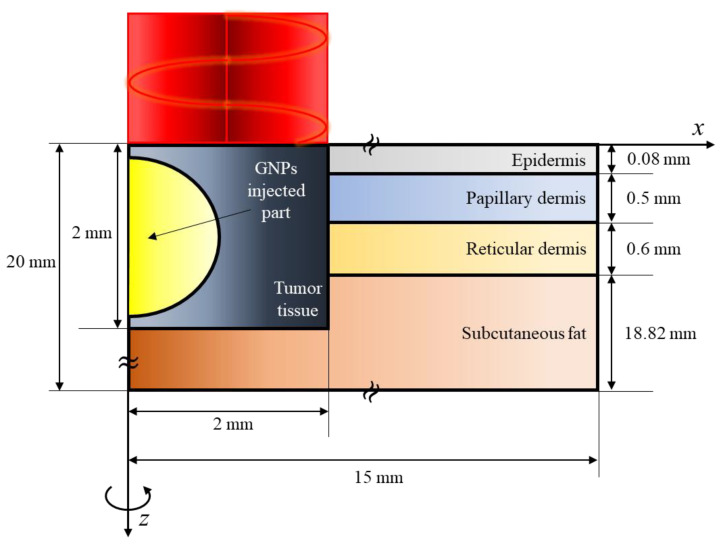
Schematic of numerical model.

**Figure 5 pharmaceutics-15-00911-f005:**
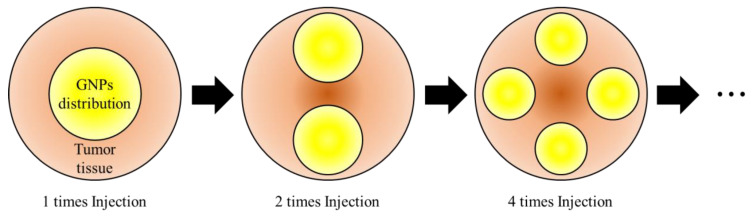
Reduction of the distribution area of GNPs as the number of injection increases.

**Figure 6 pharmaceutics-15-00911-f006:**
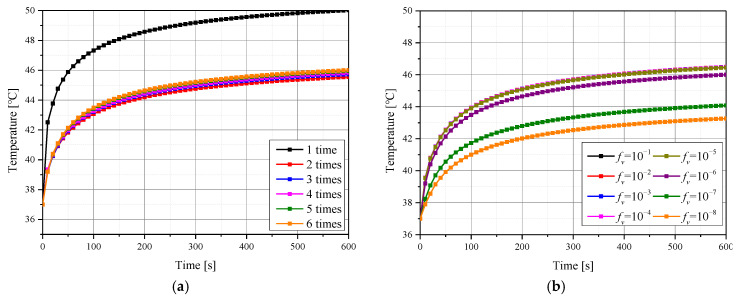
Temperature change of tumor tissue over time for: (**a**) various numbers of GNPs injections (*f_v_* = 10^−6^); (**b**) various volume fractions of GNPs (over six injections).

**Figure 7 pharmaceutics-15-00911-f007:**
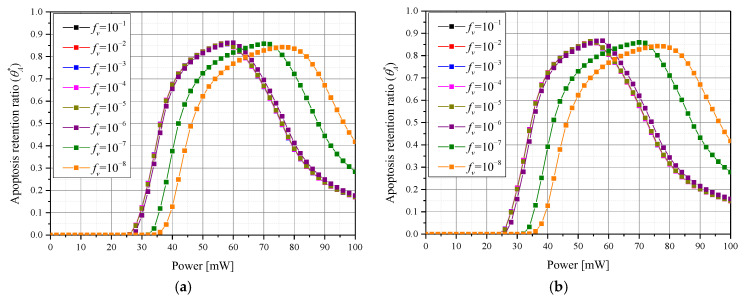
Apoptosis retention ratio ($\theta_{A}^{*}$) for various volume fraction of GNPs (*f_v_*) over (**a**) three injections and (**b**) six injections.

**Figure 8 pharmaceutics-15-00911-f008:**
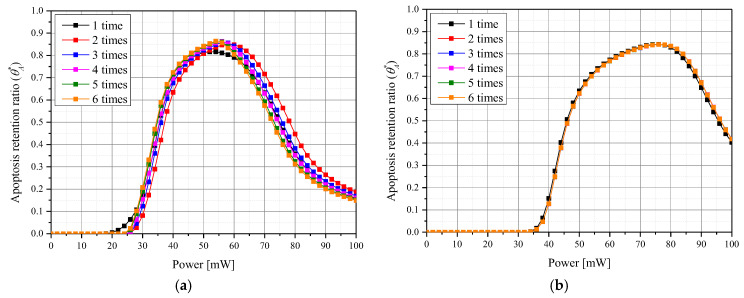
Apoptosis retention ratio ($\theta_{A}^{*}$) for various number of GNPs injections at (**a**) *f_v_
*= 10^−3^ and (**b**) *f_v_
*= 10^−8^.

**Figure 9 pharmaceutics-15-00911-f009:**
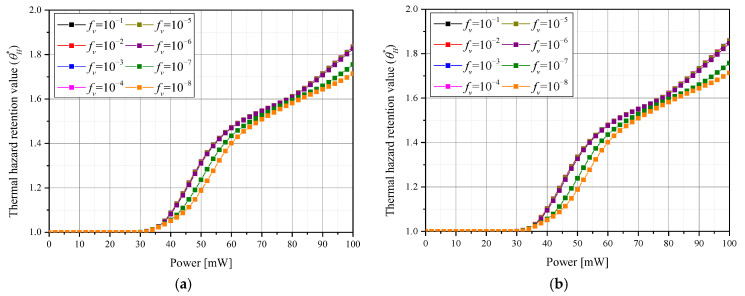
Thermal hazard retention value ($\theta_{H}^{*}$) for various volume fractions of GNPs (*f_v_*) over (**a**) three injections and (**b**) six injections.

**Figure 10 pharmaceutics-15-00911-f010:**
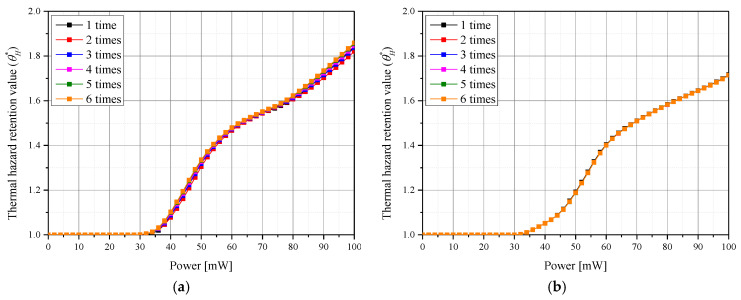
Thermal hazard retention value ($\theta_{H}^{*}$) for various numbers of GNPs injections when (**a**) *f_v_
*= 10^−3^ and (**b**) *f_v_
*= 10^−8^.

**Figure 11 pharmaceutics-15-00911-f011:**
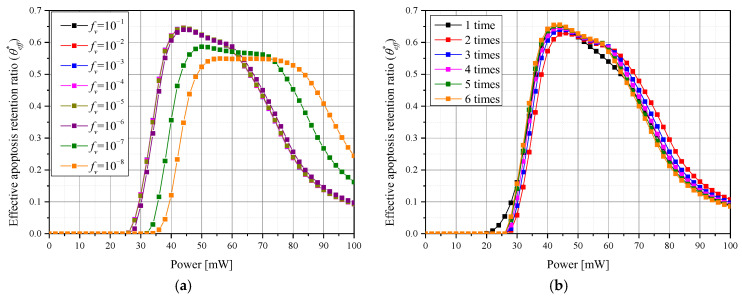
Effective apoptosis retention ratio ($\theta_{eff}^{*}$) for (**a**) various *f_v_* values over three GNPs injections; (**b**) various numbers of GNPs injections for *f_v_
*= 10^−6^.

**Table 1 pharmaceutics-15-00911-t001:** Thermal and optical properties of BALB/c mouse [42,43].

Thermal conductivity *k_m_* (W/mK)	0.34
Density ρ (kg/m^3^)	1000
Specific heat *c_v_* (J/kgK)	3000
$\mathrm{Absorption}\;\mathrm{coefficient}\;\mu_{abs}$ (1/mm)	6.1
$\mathrm{Scattering}\;\mathrm{coefficient}\;\mu_{sca}$ (1/mm)	40.65
Anisotropy factor *g*	0.8

**Table 2 pharmaceutics-15-00911-t002:** Properties of skin layer and tumor tissue [11,45,46,47,48,49,50,51].

	t (mm)	*k_m_* (W/mK)	ρ (kg/m^3^)	*c_v_* (J/kgK)	g	$\mu_{abs}$ (1/mm)	$\mu_{sca}$ (1/mm)
Epidermis	0.08	0.235	1200	3589	0.8	0.4	45
Papillary dermis	0.5	0.445	1200	3300	0.9	0.38	30
Reticular dermis	0.6	0.445	1200	3300	0.8	0.48	25
Subcutaneous fat	18.82	0.19	1000	2500	0.75	0.43	5
Tumor	2	0.495	1070	3421	0.8	0.047	0.883

**Table 3 pharmaceutics-15-00911-t003:** Parameters of numerical analysis.

Parameter	Case	Number	Remarks
Laser power (*P_l_*)	0 to 100 mW	51	Intv: 2 mW
Volume fraction of GNPs (*f_v_*)	10^−1^ to 10^−8^	8	Intv: 10^−1^
Number of injected GNPs	1 to 6	6	Intv: 1

**Table 4 pharmaceutics-15-00911-t004:** Absorption and scattering coefficients of tumor with GNPs.

f_v_	10^−1^	10^−2^	10^−3^	10^−4^	10^−5^	10^−6^	10^−7^	10^−8^
$\mu_{abs}$ (cm^−1^)	5,016,309.16	501,631.34	50,163.55	5016.78	502.10	50.63	5.48	0.97
$\mu_{sca}$ (cm^−1^)	1,059,239.60	105,931.91	10,601.14	1068.06	114.75	19.42	9.89	8.94

## Data Availability

Data sharing is not applicable to this article.

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
