# Peer review of "Quantitative Analysis of Photothermal Therapy of Tumor Tissue Using Various Gold Nanoparticle Injection Schemes"

_pharmaceutics, 2023, doi:10.3390/pharmaceutics15030911_

Round 1

Reviewer 1 Report

Please see the attached authors comments file.

Author Response

Please confirm the attached files.

Reviewer 2 Report

The manuscript try to deal with an effectivity of photothermal therapy, where the gold nanoparticles serves as light absorbers, which provides the heat effect of the light. The manuscript provides numerical simulation of the effect of repeated injected dosage and volume fraction. The manuscript seems to be providing quite usefful theoreticla predictions of these paramaters on the final apoptotic effect.

However, I have several issues to be elucidated by authors.

L137: delivery paths should comprehed also besides the i.v. injection also intraartetila route, which used for the administration of heterogeneous systems and chemotherapies. Moreovere, thorough the i.a. administration, one can benefit from the EPR effect a lot.

L141: authors deal just with the loses during the distribution after the administration, but there can be a clearence of the GDNPs also by various immunity mechanisms from the tumour environment.

Generally: the effect of the dose is related to the volume fraction of GDNPs in the tumor. However, the absorption of the light is mediated by the surface of the nanoparticles. In other words, it is a difference, when the heating is mediated with GDNPS at the same volume fraction, but with various particle diamaters. In the manuscript there is not mentioned any effect of nanoparticle size . Moreover, the positions of absorption peaks differs with GDNPs size. Therefore I recomend to author to a) define an effect of GDNPS size and b) explain, why the volume fraction is better parameter than area of particle interface.

Author Response

Please confirm the attached files.

Reviewer 3 Report

In this research, numerical modeling was used to mimic photothermal therapy combined with direct injection of gold nanoparticles into tumor tissue. The treatment impact caused by varying the laser's intensity, the volume fraction of the injected gold nanoparticles, and the number of gold nanoparticle injections was measured quantitatively. Laser absorption and scattering in tissue were identified using the Monte Carlo approach, and the optical characteristics of the total medium were calculated using the Discrete Dipole Approximation method. In addition, the treatment impact of photothermal therapy was assessed, and best practice conditions for its application were recommended, by verifying the temperature distribution of the entire medium using the predicted light absorption distribution. The future adoption of photothermal therapy is anticipated to be boosted by this.

2.2 Numerical investigation: A 1064 nm single wavelength Gaussian distribution laser with a radius of 2 mm was 250 used as the heat source. why? what is the reference for this?

It was assumed that GNPs were distributed in a spherical shape at each injection site, and the total volume fraction of GNPs in the tumor was assumed to be the same regardless of the number of injections of GNPs. what is evidence for this assumption? Is it confirmed?

The laser irradiation time was set to 600 s, and the intensity of the laser was increased in 2 mW intervals from 0–100 mW. The volume fraction of injected GNPs was increased from 10-1 10-8 at intervals of 10-1, in a total of six steps. what is reference for this procedure?

Author Response

Please confirm the attached files.

Round 2

Reviewer 2 Report

The manuscript underwent a significant improvement. However, still I would like to recommend some minor changes to be considered:

Please, check line 249, where it is written GNP diameter 10 mm. It is obviously a nonsense.

Line 300: effective radius of GNP in model was chosen as 20 nm. How it is comparable with the experiment, where were used 10 nm (written 10 mm) particles? If the meaning of "effective radius" means something different than the size in line 249, it should be defined.

It should be somehow emphasized, that in the manuscript there is no information about the behaviour of the particles after their administration. They can coagulate, conjugate with plasma proteis, or protensi of extracular matrix etc. Therefore the reality could be significantly different from the model due to that. It is also pretty clear, that uncoated particles will not acquire any permission/validation without an appropriated biocompatibilisation coating, which will also affect the game in a sense of physics.

Author Response

Please confirm the attached files.

Reviewer 3 Report

Acceptable in current form.

Author Response

Thank you for your review.